# Infant Saliva Microbiome Activity Modulates Nutritional Impacts on Neurodevelopment

**DOI:** 10.3390/microorganisms11082111

**Published:** 2023-08-18

**Authors:** Terrah Keck-Kester, Steven D. Hicks

**Affiliations:** Department of Pediatrics, Penn State College of Medicine, Hershey, PA 17033, USA

**Keywords:** neurodevelopment, microbiome, nutrition, saliva

## Abstract

Neurodevelopment is influenced by complex interactions between environmental factors, including social determinants of health (SDOH), nutrition, and even the microbiome. This longitudinal cohort study of 142 infants tested the hypothesis that microbial activity modulates the effects of nutrition on neurodevelopment. Salivary microbiome activity was measured at 6 months using RNA sequencing. Infant nutrition was assessed longitudinally with the Infant Feeding Practices survey. The primary outcome was presence/absence of neurodevelopmental delay (NDD) at 18 months on the Survey of Wellbeing in Young Children. A logistic regression model employing two microbial factors, one nutritional factor, and two SDOH accounted for 33.3% of the variance between neurodevelopmental groups (*p* < 0.001, AIC = 77.7). NDD was associated with Hispanic ethnicity (OR 18.1, 2.36–139.3; *p* = 0.003), no fish consumption (OR 10.6, 2.0–54.1; *p* = 0.003), and increased *Candidatus Gracilibacteria* activity (OR 1.43, 1.00–2.07; *p* = 0.007). Home built after 1977 (OR 0.02, 0.001–0.53; *p* = 0.004) and *Chlorobi* activity (OR 0.76, 0.62–0.93, *p* = 0.001) were associated with reduced risk of NDD. Microbial alpha diversity modulated the effect of fish consumption on NDD (*X*^2^ = 5.7, *p* = 0.017). These data suggest the benefits of fish consumption for neurodevelopment may be mediated by microbial diversity. Confirmation in a larger, randomized trial is required.

## 1. Introduction

Recent studies highlight how complex interactions of mind, body, and environment can contribute to neurodevelopmental outcomes. These outcomes include “neurotypical” and “neurodivergent”, encompassing diagnoses from autism spectrum disorder (ASD) to intellectual disability (ID) to attention deficit hyperactivity disorder (ADHD) [1].

For neurodevelopmental conditions such as autism spectrum disorder, it is well established that genetics play a crucial role [2,3]. Genetic factors are estimated to contribute between forty percent and eighty percent of an individual’s risk for autism. Although hundreds of gene variants may result in an autism phenotype, the function of these gene products tend to converge on a smaller set of physiologic pathways [2]. In addition, some gene variants may contribute to similar or overlapping symptomology in conditions such as autism and ADHD. However, emerging evidence suggests that there is a complex interplay between genetic signatures and the environment; a true connection between “nature” and “nurture”. Some important environmental factors that appear to have a critical impact on neurodevelopment include social determinants of health (SDOH), stress or adverse childhood events (ACEs), environmental toxins, nutrition, and even the microbiome. 

Early life environment, in particular, is crucial to lifelong cognitive function and social ability. For example, factors such as toxic stress and maternal obesity can lead to alterations in neurodevelopment [4]. Other important environmental factors include family income, access to healthcare, racial and ethnic bias, or exposure to pollutants and toxic chemicals. A recent review of air pollution and neurodevelopment concluded that this environmental factor may be most important in the prenatal/perinatal period, and have the greatest impact on intellectual functioning and attention [5]. Similarly, environmental toxins such as lead (found in the paint of older homes) can lead to intellectual disability if not identified and removed from a child’s environment in a timely manner [6]. Quantifying the relative contributions of such environmental toxins to child development is difficult because their impact can be amplified by the presence of other toxins and they can covary with social determinants of health [6,7]. 

Recent studies suggest the microbiome may actively contribute to neurodevelopment. Early, post-natal brain development occurs at the same time as acquisition and reorganization of microbiota in both the oropharynx (saliva) and the lower intestinal tract (stool) [8,9]. Studies by Clarke and colleagues (2013) have identified specific brain regions involved in early developmental processes that are responsive to the microbiome–gut–brain access signaling [10]. There is growing recognition that the gut–brain axis involves bidirectional communication between the central and the enteric nervous systems, linking cognitive centers of the brain with peripheral metabolites and inflammation produced by microbial communities in the gastrointestinal tract. 

Nutritional factors have also been associated with neurodevelopmental outcomes. A recent review of prenatal nutrition noted a relationship between insufficient nutrient intake during pregnancy, anatomical defects in the neonatal brain (e.g., diminished cerebral volume, spina bifida, alteration of hypothalamic and hippocampal pathways), and risk of neuropsychiatric diagnoses (e.g., ASD, ADHD, anxiety, depression), altered cognition, visual impairment, and motor deficits [11]. This includes both caloric deficient diets and high calorie, high fat, low nutrient-rich diets, such as the standard American diet [11]. 

In particular, fish intake during pregnancy has been associated with improved neurodevelopmental outcomes at 18 months of age [12,13]. However, it is well documented that deficiencies in other important vitamins and nutrients, such as iron, folic acid, and Vitamin D among others, can have lasting impacts on brain development [11].

Despite the growing body of literature involving environmental impacts on neurodevelopment, few studies have attempted to quantify the relative contributions of multiple environmental factors (i.e., social determinants, nutrition, microbiome) to child neurodevelopment. To our knowledge, no study has examined whether the microbiome modulates the effects of SDOH or nutrition on neurodevelopment. This study tested the hypothesis that child neurodevelopmental outcomes at 18 months are influenced by distinct interactions between infant nutrition and oral microbiome diversity in the first year of life. To test this hypothesis, we performed a longitudinal cohort study involving 142 term infants, followed from birth to 18 months. 

## 2. Materials and Methods

Ethical approval for this study was provided by the Institutional Review Board at the Penn State College of Medicine (STUDY00008657). All participants provided informed, written consent at the time of enrollment. The study was registered at clinicaltrials.gov (NCT04017520; accessed on 9 January 2023).

### 2.1. Participants

This was a longitudinal, prospective cohort study of term, breastfeeding infants enrolled at birth and followed through 18 months of life. Inclusion criteria were: (1) delivery > 35 weeks; (2) intention to receive well child care at a primary care clinic affiliated with our academic medical institution; and (3) intention to breastfeed for at least four months. Exclusion criteria were: (1) maternal/infant medical conditions that could impact ability to breastfeed (i.e., maternal drug use, maternal infection with human immunodeficiency virus, infant metabolic disorder); (2) maternal age < 18 years; and (3) inability to complete surveys written in English. The primary medical outcome was presence or absence of neurodevelopmental delay (NDD) on the Survey of Wellbeing in Young Children (SWYC) at 18 months [14]. The SWYC is a validated screening tool that can be used to longitudinally assess neurodevelopment, employing 10 questions regarding child motor, language, social, and cognitive development as part of its developmental domain. Recruitment occurred at the newborn nursery and outpatient pediatric clinics affiliated with our academic medical center in Hershey, PA, USA. Recruitment occurred between April 2018 and October 2021. There were 2487 infants screened for eligibility, of whom 359 met the criteria, 221 consented to participate, and 142 had neurodevelopmental assessment completed at 18 months. A power analysis determined that this sample size provided 84% power to detect a 1.5-fold difference between groups on a two-tailed non-parametric *t*-test with alpha set at 0.05.

### 2.2. Data Collection

Electronic, standardized surveys were administered by trained research staff at well child visits at enrollment, 1 month, 6 months, and 12 months after birth. Data were stored and curated using REDCap software (version 10.6.3). Infant biologic sex, race, and ethnicity were reported. SDOH were measured at one month with the Phen X Toolkit and the National Survey of Lead and Allergens in Housing (NSLAH) [15]. The SDOH recorded for this study were: maternal education (college degree versus no college degree), marital status (married versus not married), health insurance status (private insurance versus public insurance), household income (less than $25,000 versus greater than or equal to $25,000), number of persons in the household, and atmospheric pollution (present versus absent). These factors were selected based upon prior studies suggesting that they might impact child development [4,5,6,7]. Maternal nutrition was measured at enrollment with the maternal Dietary Screener Questionnaire (DSQ) [16], and infant nutrition was assessed at 6 and 12 months with the Infant Feeding Practices II Survey (IFP-II) [17]. Infant consumption of breast milk at six months was examined (no breast milk versus breast milk at least once daily). Infant consumption of nine other foods was assessed at 12 months: daily consumption of dairy, fruit, vegetables, meat/chicken; and weekly consumption of soy, juice, fish, eggs, and sweets. There were 107 infants with complete SDOH and nutritional data who were included in the analysis. No data imputation was performed. 

### 2.3. Saliva Microbiome Analysis

Microbiome transcriptional activity was measured in saliva at 6 months. We chose saliva because of its proximity to the central nervous system, its ease of access at standardized well child visits, and because the oropharynx represents one of the first sites of microbial contact for infants exploring their physical world. Saliva samples were collected with a highly absorbent swab, containing RNA stabilizer. Daytime, non-fasting samples were collected by placing the swab in the sub-lingual and parotid regions for 10–20 s, per manufacturer instructions (DNA Genotek, Ottawa, ON, Canada). Saliva was aliquoted within four weeks, and stored at −80 °C prior to RNA extraction. RNA was extracted using the miRNeasy Kit (Qiagen, Inc., Germantown, MD, USA), and RNA integrity and quantity were assessed with an Agilent Bioanalyzer 2100 (Agilent, Santa Clara, CA, USA). RNA libraries were prepared with the Illumina TruSeq Small RNA Prep protocol, as we have previously described [18]. High throughput sequencing was performed at the State University of New York Upstate Molecular Analysis Core with a NextSeq500 (New York City, NY, USA) instrument at a depth of 10 million, 50 base, paired-end reads per sample. RNA reads that were unaligned to the human genome (hg38, Bowtie2 aligner) were aligned to the NCBI RefSeq genome at the phylum level using Kraken (v1) in Partek Flow (Partek; St. Louis, MO, USA). The integrity of the RNA sequencing results was verified through read quality score and total read count. Median raw read depth (1.3 × 10^6^) was compared between groups and no difference was present (*p* = 0.33, d = −0.2). The RNA features with consistent detection (raw read counts ≥10 in 10% of samples) were sum-normalized and mean-center scaled. 

### 2.4. Statistical Analysis

Medical and demographic characteristics were compared between participants with typical development (*n* = 118) and participants with NDD (*n* = 24) using Student’s *t*-tests or chi-square tests, as appropriate. Multiple testing correction was performed using the Benjamini Hochberg method. A feed-backward logistic regression model was used to identify factors that contributed significantly (*p* < 0.05) to the probability of typical development at 18 months. The following categories of factors were assessed: participant characteristics (biologic sex, race, ethnicity); SDOH (maternal education, marital status, health insurance, home age, household income, persons in household, atmospheric pollution); nutrition (breastfeeding duration, dairy, soy, juice, fruit, vegetables, meat/chicken, fish, eggs, sweets); and salivary microbes (Simpson Alpha Diversity and individual phyla with reliable detection, defined by median read count ≥ 2). Factor significance was determined with omnibus analysis of variance (ANOVA) testing and odds ratios were reported. All factors were assessed for collinearity and overall model fit was evaluated with Akaike Information Criterion (AIC). Interactions between significant factors were explored. Statistical analysis was completed in Jamovi (v2.3) software. 

## 3. Results

### 3.1. Participants

Of the 142 children who completed the study, the majority were female (67, 54%), White (102, 72%), and non-Hispanic (119, 83%) (Table 1). Most mothers were married, with a college degree and private health insurance. Less than half of participating infants lived in a house built before 1977 (when lead paint was banned), had a household income <$25,000, or were exposed to atmospheric pollution. There were 24 (16%) children at risk for NDD based on their 18 month SWYC score. Average SWYC score for the NDD group was 6.9 (±2), compared with 13.7 (±3) for the group with typical neurodevelopment (ND). There were no participant characteristics or SDOH that differed (*p* < 0.05) between children with typical ND and peers with NDD. 

### 3.2. Oral Microbial Activity

Transcripts from 17 of the 38 phyla (44%) were reliably detected in infant saliva at 6 months of age (Figure 1). The most transcriptionally active phyla were Firmicutes (median normalized read count: 108,560, range: 22,741–483,921), Bacteroidetes (median: 8296, range: 86–62,226), Proteobacteria (median: 4855, range: 345–28,888), and Actinobacteria (median: 4157, range: 196–54,020). Together, these four phyla accounted for 98% of all microbial transcripts in infant saliva. Total read count did not differ between children with typical ND and peers with NDD (*p* > 0.05). Microbial diversity (measured via Simpson Index) was also similar between children with typical ND (0.86 ± 0.11) and peers with NDD (0.88 ± 0.11; *p* = 0.30). Cyanobacteria activity was nominally higher among children with typical ND compared with peers with NDD; however, this trend did not survive multiple testing correction (d = 0.35, *p* = 0.012, adj *p* = 0.21). No other phyla differed between groups.

### 3.3. Nutrition

The majority of participants consumed human milk at least once per day for six months or more (106/142, 74%) (Table 2). At 12 months of age, few consumed soy (12/107, 11%) or juice (34/107, 31%) weekly, but most consumed vegetables (86/107, 80%) or fruit (85/107, 79%) daily. Nearly half consumed sweets (50/107, 46%) or fish (51/107, 47%) weekly. Nearly one third consumed dairy (28/107, 26%) or meat/chicken (42/107, 39%) daily. Children with NDD at 18 months were less likely to consume fish weekly at 12 months, compared to peers with typical ND (X^2^ = 4.22, *p* = 0.040).

### 3.4. Modeling the Effects of Nutrition, SDOH, and Microbial Activity on ND

A feed-backward logistic regression approach was used to model the contributions of nutrition, SDOH, and microbial activity to ND outcomes. The final model (retaining factors with *p* < 0.05 on Omnibus ANOVA) employed two microbial factors, one nutritional factor, and two SDOH, while accounting for 33.3% of the variance between ND groups (*p* < 0.001, AIC = 77.7) (Figure 2). NDD was associated with Hispanic ethnicity (OR: 18.1, 2.36–139.3; *p* = 0.003), no infant fish consumption at 12 months (OR: 10.6, 2.0–54.1; *p* = 0.003), and increased *Candidatus Gracilibacteria* activity (OR: 1.43, 1.00–2.07; *p* = 0.007), whereas home built after 1977 (OR: 0.02, 0.001–0.53; *p* = 0.004) and *Chlorobi* activity (OR: 0.76, 0.62–0.93, *p* = 0.001) were associated with reduced risk of NDD. Microbial alpha diversity modulated the effect of fish consumption on neurodevelopmental outcomes (X^2^ = 5.7, *p* = 0.017) (Figure 3).

## 4. Discussion

This prospective cohort study of 142 term infants identified individual traits, social determinants of health, nutritional elements, and microbial factors that contribute to risk of NDD at 18 months. Specifically, we found that Hispanic ethnicity, age of home, fish consumption at 12 months, and activity of two salivary microbes (*Candidatus Gracilibacteria* and *Chlorobi*) at 6 months were related to ND outcomes. Intriguingly, the effect of fish intake on ND was modulated by bacterial diversity in the oropharynx. ND outcomes (measured by the SWYC) improved with increasing microbial diversity for children who consumed fish at least once per week.

Previous studies have suggested that fish intake may have positive impacts on child brain development and ND outcomes as a result of long chain polyunsaturated fatty acids (LC-PUFA) [19,20]. However, there are conflicting findings regarding this relationship [21]. In addition, studies of the gut microbiome have found potential relationships between microbial diversity and neurodevelopment [22], as well as child social skills [23]. To our knowledge, this study is among the first to demonstrate that these two factors (fish consumption and microbial diversity) may *interact* to impact child ND. 

Few prior studies have noted relationships between human health outcomes and abundance/activity of *Chlorobi* or *Candidatus Gracilibacteria* [24,25]. Both phyla inhabit aqueous, oceanic environments but are not typically considered human pathogens. Therefore, we posit that these phyla represent harbingers of microbial diversity and specifically represent microbial populations that could aid metabolism of fish-related nutritional components [26]. For example, *Candidatus Gracilibacteria* has been shown to break down oil products and could therefore play a role in the metabolism of LC-PUFAs [26].

Ultimately, the impacts of nutrition and the microbiome on human health cannot be evaluated in a vacuum. Social determinants of health and prenatal/perinatal environment have been shown to have profound effects on child neurodevelopment [11,27,28]. In this cohort, home age and ethnic minority status were associated with increased odds of NDD at 18 months. Prior research has shown that racial/ethnic minorities face health inequities that portend poorer long-term neurodevelopmental outcomes [29]. This may result from limited English proficiency, systemic racial bias within healthcare, or socioeconomic disadvantages [30]. The current study controlled for English proficiency (through inclusion/exclusion criteria) and socioeconomic status (using income and health insurance as covariates). It is possible that home age also represents a socioeconomic measure. In this Central Pennsylvania community, gentrification of older homes is a common practice. Thus, older homes, which were paradoxically associated with decreased odds of NDD, may reflect socioeconomic means, rather than exposure to lead-based paint. 

Understanding the relative contributions of SDOH, nutrition, and the microbiome to ND outcomes holds enormous clinical importance for families, physicians, and society at large. It has been projected that the societal costs of autism spectrum disorder will reach $461 billion in the United States by 2025 [31]. The current clinical paradigm for autism and other conditions involving NDD involves regular, routine screening [32]. However, many families face barriers to routine pediatric care, many practices fail to employ the recommended screening regimen, and recent studies have called into question the accuracy and efficacy of screening tools [33,34]. Expanding our understanding of early life factors that impact developmental trajectories could provide a unique opportunity to shift from a detection and intervention paradigm to one of proactive prevention. For example, if the current findings were to be replicated in a large prospective study, a simple intervention involving more extensive prenatal and postnatal nutrition counseling, and early introduction of fish consumption could be assessed for neurodevelopmental impact, and possibly targeted probiotics. 

We note, however, that the preliminary results in this study should be interpreted with caution. The study’s sample size provides limited power to interrogate multi-factorial relationships. As a result, microbial analysis was limited to 38 phyla. Deeper relationships may exist at the genus or species levels. Although rates of enrollment and study completion were consistent with many longitudinal cohort studies of children, missing data for the IFP-II (107/142, 75%) and microbial activity (121/142, 85%) may introduce bias from data not missing at random. Our assessment of neurodevelopment with a standardized survey (SWYC) promotes reproducibility; however, we note that this screening tool is not a definitive measure of NDD. Future studies should employ clinician assessments of ND and examine the effects of microbial/nutrition factors on specific sub-scales (i.e., gross motor, fine motor, socialization, cognition). Finally, to assess causation, these findings require replication in a randomized control trial, assigning families to fish consumption groups while longitudinally monitoring changes in the microbiome over time. Assessment of fecal microbe abundance using 16S approaches may also yield interesting and complementary information to measures of oral microbiome activity. 

## 5. Conclusions

This study demonstrates that child ND may be impacted by complex interactions between nutrition and the microbiome. Specifically, oral microbial diversity at six months modulates the impact of infant fish consumption on ND outcomes at 18 months. This relationship exists even after controlling for SDOH such as ethnic minority status and age of home. We posit that microbial diversity may be important for metabolism and utilization of essential nutrients, such as LC-PUFA, associated with fish consumption. Additional research is required to confirm these findings.

## Figures and Tables

**Figure 1 microorganisms-11-02111-f001:**
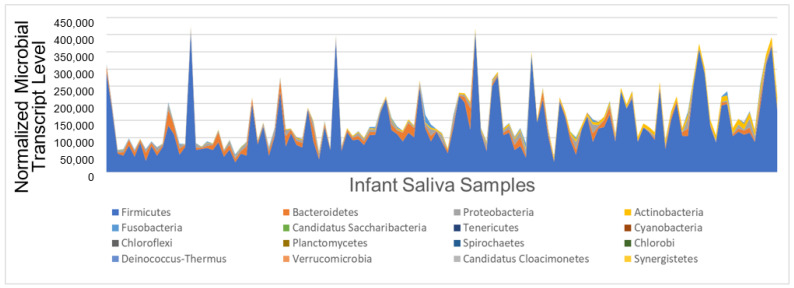
Salivary microbe activity at six months. The stacked area chart displays sum normalized microbial read counts (*y*-axis) across 121 infant saliva samples (*x*-axis). Read counts for the 17 microbial phyla with reliable detection in infant saliva at six months of age are shown. Four phyla (*Firmicutes*, dark blue; *Bacteroidetes*, orange; *Proteobacteria*, grey; and *Actinobacteria*, yellow) accounted for 98% of microbial transcripts across all samples. However, the samples display heterogeneity in total microbial transcript abundance and the relative contributions of the four most abundant phyla. *Cyanobacteria* transcripts were nominally higher among children with typical neurodevelopment (d = 0.35, *p* = 0.012, adj *p* = 0.21) but accounted for only 0.01% of microbial transcripts.

**Figure 2 microorganisms-11-02111-f002:**
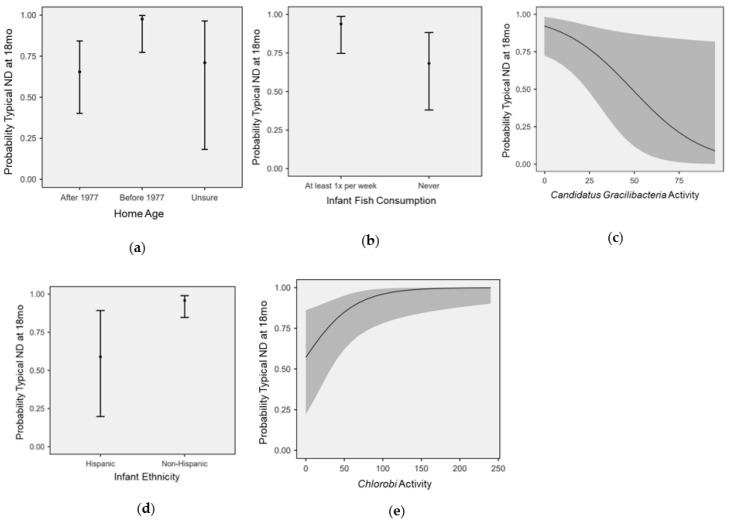
Factors associated with neurodevelopment at 18 months. The marginal means plots display the relationship between probability of typical neurodevelopment (ND) at 18 months and 5 factors that were significantly related on logistic regression: infant ethnicity (**a**) (OR 18.1, 2.36–139.3; *p* = 0.003), fish intake (**b**) (OR 10.6, 2.0–54.1; *p* = 0.003), *Candidatus Gracilibacteria* (**c**) (OR 1.43, 1.00–2.07; *p* = 0.007), age of home (**d**) (OR 0.02, 0.001–0.53; *p* = 0.004), and *Chlorobi* (**e**) (OR 0.76, 0.62–0.93, *p* = 0.001). Grey shaded area denotes 95% confidence interval (**c**,**e**).

**Figure 3 microorganisms-11-02111-f003:**
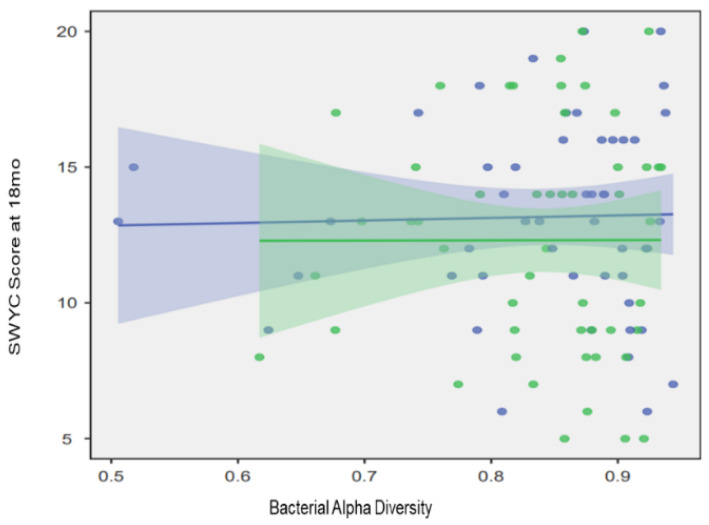
Bacterial diversity modulates the effect of fish intake on child neurodevelopment. The scatter plot displays the relationship between fish intake and neurodevelopment at 18 months (mo), measured with the Survey of Wellbeing in Young Children (SWYC). This relationship was modulated by microbial diversity (*X*^2^ = 5.7, *p* = 0.017) on regression analysis. Blue represents children with fish intake at least once per week, whereas green represents children without fish consumption. Trend lines with 95% confidence intervals are displayed for each group.

**Table 1 microorganisms-11-02111-t001:** Participant characteristics and social determinants of health.

Characteristics and SDOH	All N = 142	Typical ND N = 118	NDD N = 24
Female sex, *n* (%)	77 (54)	65 (55)	12 (50)
White race, *n* (%)	102 (71)	86 (72)	16 (66)
Hispanic ethnicity, *n* (%)	23 (16)	17 (14)	6 (25)
Maternal college education, *n* (%)	120 (84)	99 (83)	21 (87)
Married, *n* (%)	109 (76)	90 (76)	19 (79)
Private health insurance, *n* (%)	114 (80)	92 (77)	22 (91)
Home built before 1977, *n* (%)	28 (19)	27 (22)	1 (4)
Household income < $25,000, *n* (%)	12 (8)	9 (7)	3 (12)
Persons in household, average (range)	3.9 (2–9)	3.9 (2–9)	3.8 (3–5)
Atmospheric pollution, *n* (%)	38 (26)	33 (27)	5 (20)

Abbreviations: neurodevelopment (ND); neurodevelopmental delay (NDD); social determinants of health (SDOH).

**Table 2 microorganisms-11-02111-t002:** Nutritional factors.

Infant Nutrition	All N = 142	Typical ND N = 118	NDD N = 24
Breastfeeding at 6 months, *n* (%)	106 (74)	90 (76)	16 (66)
Dairy ≥ once daily, *n* (%)	28 (26)	24 (27)	4 (21)
Soy ≥ once weekly, *n* (%)	12 (11)	11 (12)	1 (5)
Juice ≥ once weekly, *n* (%)	34 (31)	28 (31)	6 (31)
Fruit ≥ once daily, *n* (%)	85 (79)	70 (79)	14 (73)
Vegetables ≥ once daily, *n* (%)	86 (80)	73 (82)	13 (68)
Meat/chicken ≥ once daily, *n* (%)	42 (39)	35 (39)	7 (36)
Fish ≥ once weekly, *n* (%)	51 (47)	46 (52)	5 (26) *
Eggs ≥ once weekly, average (range)	89 (83)	76 (88)	13 (68)
Sweets ≥ once weekly, *n* (%)	50 (46)	41 (46)	9 (47)

* *p* < 0.05 on chi-square testing. Abbreviations: neurodevelopment (ND); neurodevelopmental delay (NDD). Items assessed as part of the Infant Feeding Practices—II survey were available for only 107 infants at 12 months of age (19 with NDD and 88 with typical ND).

## Data Availability

FASTQ files from RNA sequencing involved in this project have been deposited into the NIH GEO database (GSE192543). GEO repository link: https://www.ncbi.nlm.nih.gov/geo/query/acc.cgi?acc=GSE192543 (accessed on 25 December 2022).

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
