# Peer review of "Infant Saliva Microbiome Activity Modulates Nutritional Impacts on Neurodevelopment"

_microorganisms, 2023, doi:10.3390/microorganisms11082111_

Round 1
Reviewer 1 Report
This cohort study validated the relationship between infant saliva microbiome activity and neurodevelopment, which is an interesting topic. Here I have a concern. As for the detection or analyze in neurodevelopment capacity, the author only valued the time in 18 month, why only chosen this time point ? even longer time for evaluation of the neurodevelopment capacity might be needed?
Reviewer 2 Report
In this manuscript, the Authors report the impact of nutrition and microbiome on child neurodevelopment. Interestingly, Authors showed that oral microbial diversity at six months was able to influence the impact of infant fish consumption on neurodevelopmental outcomes at 18 months. The study is very interesting, well written and well conducted.
However, I have some minor concerns:
- I would suggest to better clarify inclusion and exclusion criteria. Maybe it would be useful to make a numbered list;
- It would be useful to rearrange Table 1 in order to have everything reported, such as males number, public health insurance, etc.. It is more exhaustive include such informations in table, rather than in the text;
- The graph in Figure 1 is very difficult to understand. I would suggest to show it differently, in order to understand all the different microbes;
- In Matherial and Methods and Results sections, it is stated that total participants were 142, while in the abstract is reported 221. Please unify this discrepancy.
